# One-Step Diffusion Distillation via Deep Equilibrium Models

**Zhengyang Geng**[*]
Carnegie Mellon University
zgeng2@cs.cmu.edu

**Ashwini Pokle**[*]
Carnegie Mellon University
apokle@cs.cmu.edu

**J. Zico Kolter**
Carnegie Mellon University
Bosch Center for AI
zkolter@cs.cmu.edu

## Abstract

Diffusion models excel at producing high-quality samples but naively require hundreds of iterations, prompting multiple attempts to distill the generation process into a faster network. However, many existing approaches suffer from a variety of challenges: the process for distillation training can be complex, often requiring multiple training stages, and the resulting models perform poorly when utilized in single-step generative applications. In this paper, we introduce a simple yet effective means of distilling diffusion models *directly* from initial noise to the resulting image. Of particular importance to our approach is to leverage a new Deep Equilibrium (DEQ) model as the distilled architecture: the Generative Equilibrium Transformer (GET). Our method enables fully offline training with just noise/image pairs from the diffusion model while achieving superior performance compared to existing one-step methods on comparable training budgets. We demonstrate that the DEQ architecture is crucial to this capability, as GET matches a $5\times$ larger ViT in terms of FID scores while striking a critical balance of computational cost and image quality. Code, checkpoints, and datasets are available here.

## 1 Introduction

Diffusion models [35, 93, 95, 96] have demonstrated remarkable performance on a wide range of generative tasks such as high-quality image generation [71, 82, 85, 87] and manipulation [19, 67, 71, 85, 86], audio synthesis [39, 40, 51, 58], video [36, 92], 3D shape [44, 78], text [30, 56], and molecule generation [18, 38, 107]. These models are trained with a denoising objective derived from score matching [42, 95, 96], variational inference [35, 49, 93], or optimal transport [57, 60], enabling them to generate clean data samples by progressively denoising the initial Gaussian noise during the inference process. Unlike adversarial training, the denoising objective leads to a more stable training procedure, which in turn allows diffusion models to scale up effectively [82, 85, 87]. Despite the promising results, one major drawback of diffusion models is their slow generative process, which often necessitates hundreds to thousands of model evaluations [22, 35, 96]. This computational complexity limits the applicability of diffusion models in real-time or resource-constrained scenarios.

In an effort to speed up the slow generative process of diffusion models, researchers have proposed distillation methods [11, 68, 88, 97, 111] aimed at distilling the multi-step sampling process into a more efficient few-step or single-step process. However, these techniques often come with their

---

[*]Equal Contribution. Correspondence to Zhengyang Geng and Ashwini Pokle

37th Conference on Neural Information Processing Systems (NeurIPS 2023).

own set of challenges. The distillation targets must be carefully designed to successfully transfer knowledge from the larger model to the smaller one. Further, distilling a long sampling process into a few-step process often calls for multiple training passes. Most of the prevalent techniques for online distillation require maintaining dual copies of the model, leading to increased memory and computing requirements. As a result, there is a clear need for simpler and more efficient approaches that address the computational demands of distilling diffusion models without sacrificing the generative capabilities.

In this work, our objective is to streamline the distillation of diffusion models while retaining the perceptual quality of the images generated by the original model. To this end, we introduce a simple and effective technique that distills a multi-step diffusion process into a single-step generative model, using solely noise/image pairs. At the heart of our technique is the Generative Equilibrium Transformer (GET), a novel Deep Equilibrium (DEQ) model [5] inspired by the Vision Transformer (ViT) [25, 75]. GET can be interpreted as an infinite depth network using weight-tied transformer layers, which solve for a fixed point in the forward pass. This architectural choice allows for the adaptive application of these layers in the forward pass, striking a balance between inference speed and sample quality. Furthermore, we incorporate an almost parameter-free class conditioning mechanism in the architecture, expanding its utility to class-conditional image generation.

Our direct approach for distillation via noise/image pairs generated by a diffusion model, can, in fact, be applied to both ViT and GET architectures. Yet, in our experiments, we show that the GET architecture, in particular, is able to achieve substantially better quality results with smaller models. Indeed, GET delivers perceptual image quality on par with or superior to other complex distillation techniques, such as progressive distillation [68, 88], in the context of both conditional and unconditional image generation. This leads us to explore the potential of GETs further. We preliminarily investigate the scaling law of GETs—how its performance evolves as model complexity, in terms of parameters and computations, increases. Notably, GET exhibits significantly better parameter and data efficiency compared to architectures like ViT, as GET matches the FID scores of a $5\times$ larger ViT, underscoring the transformative potential of GET in enhancing the efficiency of generative models.

To summarize, we make the following key contributions:

- We propose Generative Equilibrium Transformer (GET), a deep equilibrium vision transformer that is well-suited for *single-step* generative models.

- We streamline the diffusion distillation by training GET directly on noise/image pairs sampled from diffusion models, which turns out to be a simple yet effective strategy for producing one-step generative models in both unconditional and class-conditional cases.

- For the first time, we show that implicit models for generative tasks can outperform classic networks in terms of performance, model size, model compute, training memory, and speed.

## 2 Preliminaries

**Deep Equilibrium Models.** Deep equilibrium models [5] compute internal representations by solving for a fixed point in their forward pass. Specifically, consider a deep feedforward model with $L$ layers:

$$\mathbf{z}^{[i+1]} = f_\theta^{[i]}(\mathbf{z}^{[i]}; \mathbf{x}) \quad \text{for } i = 0, ..., L-1 \tag{1}$$

where $\mathbf{x} \in \mathbb{R}^{n_x}$ is the input injection, $\mathbf{z}^{[i]} \in \mathbb{R}^{n_z}$ is the hidden state of $i^{th}$ layer, and $f_\theta^{[i]} : \mathbb{R}^{n_x \times n_z} \mapsto \mathbb{R}^{n_z}$ is the feature transformation of $i^{th}$ layer, parametrized by $\theta$. If the above model is weight-tied, *i.e.*, $f_\theta^{[i]} = f_\theta, \forall i$, then in the limit of infinite depth, the output $\mathbf{z}^{[i]}$ of this network approaches a fixed point $\mathbf{z}^\star$:

$$\lim_{i \to \infty} f_\theta(\mathbf{z}^{[i]}; \mathbf{x}) = f_\theta(\mathbf{z}^\star; \mathbf{x}) = \mathbf{z}^\star \tag{2}$$

Deep equilibrium (DEQ) models [5] directly solve for this fixed point $\mathbf{z}^\star$ using black-box root finding algorithms like Broyden's method [14], or Anderson acceleration [1] in the forward pass. DEQs utilize implicit differentiation to differentiate through the fixed point analytically. Let $g_\theta(\mathbf{z}^\star; \mathbf{x}) = f_\theta(\mathbf{z}^\star; \mathbf{x}) - \mathbf{z}^\star$, then the Jacobian of $\mathbf{z}^\star$ with respect to the model weights $\theta$ is given by

$$\frac{\partial \mathbf{z}^\star}{\partial \theta} = -\left(\frac{\partial g_\theta(\mathbf{z}^\star, \mathbf{x})}{\partial \mathbf{z}^\star}\right)^{-1} \frac{\partial f_\theta(\mathbf{z}^\star; \mathbf{x})}{\partial \theta} \tag{3}$$

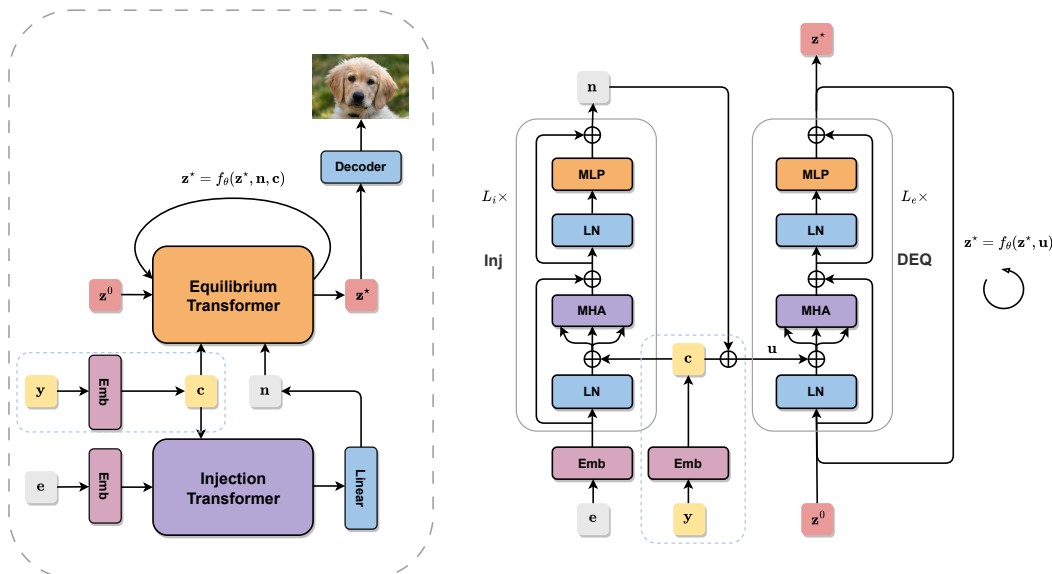

Figure 1: **Generative Equilibrium Transformer (GET).** (*Left*) GET consists of two major components: Injection transformer and Equilibrium transformer. The Injection transformer transforms noise embeddings into an input injection for the Equilibrium transformer. The Equilibrium transformer is the equilibrium layer that takes in noise input injection and an optional class embedding and solves for the fixed point. (*Right*) Details of transformer blocks in the Injection transformer (**Inj**) and Equilibrium transformer (**DEQ**), respectively. Blue dotted boxes denote optional class label inputs.

Computing the inverse of Jacobian can quickly become intractable as we deal with high-dimensional feature maps. One can replace the inverse-Jacobian term with cheap approximations [27–29] without sacrificing the final performance.

**Diffusion Models.** Diffusion models [22, 35, 93, 94] or score-based generative models [95, 96] progressively perturb images with an increasing amount of Gaussian noise and then reverse this process through sequential denoising to generate images. Specifically, consider a dataset of i.i.d. samples $p_{\text{data}}$, then the diffusion process $\{\mathbf{x}(t)\}_{t=0}^{T}$ for $t \in [0, T]$ is given by an Itô SDE [96]:

$$\mathrm{d}\mathbf{x} = \mathbf{f}(\mathbf{x}, t)\mathrm{d}t + g(t)\mathrm{d}\mathbf{w} \tag{4}$$

where $\mathbf{w}$ is the standard Wiener process, $\mathbf{f}(\cdot, t) : \mathbb{R}^d \to \mathbb{R}^d$ is the drifting coefficient, $g(\cdot) : \mathbb{R} \to \mathbb{R}$ is the diffusion coefficient, and $\mathbf{x}(0) \sim p_{\text{data}}$ and $\mathbf{x}(T) \sim \mathcal{N}(0, I)$. All diffusion processes have a corresponding deterministic process known as the probability flow ODE (PF-ODE) [96] whose trajectories share the same marginal probability densities as the SDE. This ODE can be written as:

$$\mathrm{d}\mathbf{x} = -\dot{\sigma}(t)\sigma(t)\nabla_{\mathbf{x}} \log p(\mathbf{x}, \sigma(t))\mathrm{d}t \tag{5}$$

where $\sigma(t)$ is the noise schedule of diffusion process, and $\nabla_{\mathbf{x}} \log p(\mathbf{x}, \sigma(t))$ represents the score function. Karras et al. [48] show that the optimal choice of $\sigma(t)$ in Eq. (5) is $\sigma(t) = t$. Thus, the PF-ODE can be simplified to $\mathrm{d}\mathbf{x}/\mathrm{d}t = -t\nabla_{\mathbf{x}} \log p(\mathbf{x}, \sigma(t)) = (\mathbf{x} - D_\theta(\mathbf{x}; t))/t$, where $D_\theta(\cdot, t)$ is a denoiser function parametrized with a neural network that minimizes the expected $L_2$ denoising error for samples drawn from $p_{\text{data}}$. Samples can be efficiently generated from this ODE through numerical methods like Euler's method, Runge-Kutta method, and Heun's second-order solver [3].

## 3 Generative Equilibrium Transformer

We introduce the Generative Equilibrium Transformer (GET), a Deep Equilibrium (DEQ) vision transformer designed to distill diffusion models into generative models that are capable of rapidly sampling images using only a single model evaluation. Our approach builds upon the key components and best practices of the classic transformer [99], the Vision transformer (ViT) [25], and the Diffusion transformer (DiT) [75]. We will now describe each component of the GET in detail.

**GET.** Generative Equilibrium Transformer (GET) directly maps Gaussian noises $\mathbf{e}$ and optional class labels $\mathbf{y}$ to images $\tilde{\mathbf{x}}$. The major components of GET include an injection transformer (InjectionT, Eq. (7)) and an equilibrium transformer (EquilibriumT, Eq. (8)). The InjectionT transforms tokenized noise embedding $\mathbf{h}$ to an intermediate representation $\mathbf{n}$ that serves as the input injection for the equilibrium transformer. The EquilibriumT, which is the equilibrium layer, solves for the fixed point $\mathbf{z}^\star$ by taking in the noise injection $\mathbf{n}$ and an optional class embedding $\mathbf{c}$. Finally, this fixed point $\mathbf{z}^\star$ is decoded and rearranged to generate an image sample $\tilde{\mathbf{x}}$ (Eq. (9)). Figure 1 provides an overview of the GET architecture. Note that because we are directly distilling the entire generative process, there is no need for a time embedding $t$ as is common in standard diffusion models.

$$\mathbf{h}, \mathbf{c} = \mathrm{Emb}\left(\mathbf{e}\right), \ \mathrm{Emb}\left(\mathbf{y}\right); \ \mathrm{if} \ \mathbf{y} \notin \emptyset \tag{6}$$

$$\mathbf{n} = \mathrm{InjectionT}\left(\mathbf{h}, \mathbf{c}\right) \tag{7}$$

$$\mathbf{z}^\star = \mathrm{EquilibriumT}\left(\mathbf{z}^\star, \mathbf{n}, \mathbf{c}\right) \tag{8}$$

$$\tilde{\mathbf{x}} = \mathrm{Decoder}\left(\mathbf{z}^\star\right) \tag{9}$$

**Noise Embedding.** GET first converts an input noise $\mathbf{e} \in \mathbb{R}^{H \times W \times C}$ into a sequence of 2D patches $\mathbf{p} \in \mathbb{R}^{N \times (P^2 \cdot C)}$, where $C$ is the number of channels, $P$ is the size of patch, $H$ and $W$ denotes height and width of the original image, and $N = HW/P^2$ is the resulting number of patches. Let $D = P^2 \cdot C$ denote the width of the network. We follow ViT to use a linear layer to project the $N$ patches to $D$ dimensional embedding. We add standard sinusoidal position encoding [99] to produce the noise embedding $\mathbf{h}$. Position encoding plays a crucial role in capturing the spatial structure of patches by encoding their relative positional information.

**InjectionT & EquilibriumT.** Both InjectionT and EquilibriumT are composed of a sequence of Transformer blocks. InjectionT is called only once to produce the noise injection $\mathbf{n}$, while EquilibriumT defines the function $f_\theta$ of the implicit layer $\mathbf{z}^\star = f_\theta(\mathbf{z}^\star, \mathbf{n}, \mathbf{c})$ that is called multiple times—creating a weight-tied computational graph—until convergence. A linear layer is added at the end of InjectionT to compute the noise injection $\mathbf{n}_l \in \mathbb{R}^{N \times 3D}$, $l \in [L_e]$, for each of the $L_e$ GET blocks in EquilibriumT. For convenience, we overload the notation $\mathbf{n}_l$ and $\mathbf{n}$, in the subsequent paragraphs.

**Transformer Block.** GET utilizes a near-identical block design for the noise injection (InjectionT) and the equilibrium layer (EquilibriumT), differing only at the injection interface. Specifically, the transformer block is built upon the standard Pre-LN transformer block [25, 75, 106], as shown below:

$$\mathbf{z} = \mathbf{z} + \mathrm{Attention}\left(\mathrm{LN}\left(\mathbf{z}\right), \mathbf{u}\right)$$

$$\mathbf{z} = \mathbf{z} + \mathrm{FFN}\left(\mathrm{LN}\left(\mathbf{z}\right)\right)$$

Here, $\mathbf{z} \in \mathbb{R}^{N \times D}$ represents the latent token, $\mathbf{u} \in \mathbb{R}^{N \times 3D}$ is the input injection, LN, FFN, and Attention stand for Layer Normalization [4], a 2-layer Feed-Forward Network with a hidden dimension of size $D \times E$, and an attention [99] layer with an injection interface, respectively.

For blocks in the injection transformer, $\mathbf{u}$ is equal to the class embedding token $\mathbf{c} \in \mathbb{R}^{1 \times 3D}$ for conditional image generation, i.e., $\mathbf{u} = \mathbf{c}$ for conditional models, and $\mathbf{u} = \mathbf{0}$ otherwise. In contrast, for blocks in the equilibrium transformer, $\mathbf{u}$ is the broadcast sum of noise injection $\mathbf{n} \in \mathbb{R}^{N \times 3D}$ and class embedding token $\mathbf{c} \in \mathbb{R}^{1 \times 3D}$, i.e., $\mathbf{u} = \mathbf{n} + \mathbf{c}$ for conditional models and $\mathbf{u} = \mathbf{n}$ otherwise.

We modify the standard transformer attention layer to incorporate an additive injection interface before the query $\mathbf{q} \in \mathbb{R}^{N \times D}$, key $\mathbf{k} \in \mathbb{R}^{N \times D}$, and value $\mathbf{v} \in \mathbb{R}^{N \times D}$,

$$\mathbf{q}, \mathbf{k}, \mathbf{v} = \mathbf{z}\mathbf{W}_i + \mathbf{u}$$

$$\mathbf{z} = \mathrm{MHA}\left(\mathbf{q}, \mathbf{k}, \mathbf{v}\right)$$

$$\mathbf{z} = \mathbf{z}\mathbf{W}_o$$

where $\mathbf{W}_i \in \mathbb{R}^{D \times 3D}$, $\mathbf{W}_o \in \mathbb{R}^{D \times D}$. The injection interface enables interactions between the latent tokens and the input injection in the multi-head dot-product attention (MHA) operation,

$$\mathbf{q}\mathbf{k}^\top = (\mathbf{z}\mathbf{W}_q + \mathbf{u}_q)(\mathbf{z}\mathbf{W}_k + \mathbf{u}_k)^\top = \mathbf{z}\mathbf{W}_q\mathbf{W}_k^\top\mathbf{z}^\top + \mathbf{z}\mathbf{W}_q\mathbf{u}_k^\top + \mathbf{u}_q\mathbf{W}_k^\top\mathbf{z}^\top + \mathbf{u}_q^\top\mathbf{u}_k, \tag{10}$$

where $\mathbf{W}_q, \mathbf{W}_k \in \mathbb{R}^{D \times D}$ are slices from $\mathbf{W}_i$, and $\mathbf{u}_q, \mathbf{u}_k \in \mathbb{R}^{N \times D}$ are slices from $\mathbf{u}$. This scheme adds no more computational cost compared to the standard MHA operation, yet it achieves a similar effect as cross-attention and offers good stability during training.

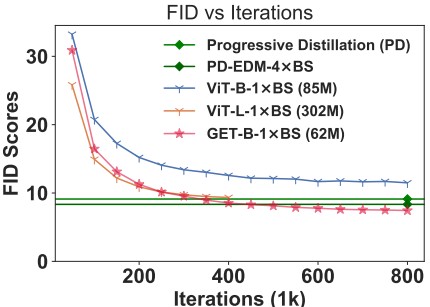
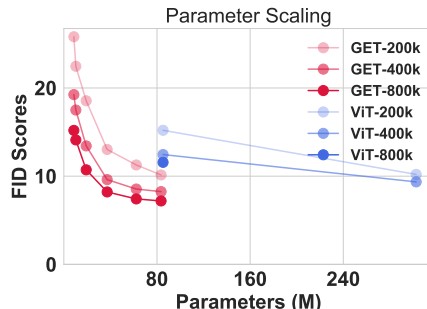

Figure 2: **Data and Parameter Efficiency of GET**:(a) (*Left*) GET outperforms PD and a 5× larger ViT in fewer iterations, yielding better FID scores. Additionally, longer training times lead to improved FID scores. (b) (*Right*) Smaller GETs can achieve better FID scores than larger ViTs, demonstrating DEQ's parameter efficiency. Each curve in this plot connects models of different sizes within the same model family at identical training iterations, as indicated by the numbers after the model names in the legend.

**Image Decoder.**    The output of the GET-DEQ is first normalized with Layer Normalization [4]. The normalized output is then passed through another linear layer to generate patches $\bar{\mathbf{p}} \in \mathbb{R}^{N \times D}$. The resulting patches $\bar{\mathbf{p}}$ are rearranged back to the resolution of the input noise $\mathbf{e}$ to produce the image sample $\tilde{\mathbf{x}} \in \mathbb{R}^{H \times W \times C}$. Thus, the decoder maps the features back to the image space.

# 4 Experiments

We evaluate the effectiveness of our proposed Generative Equilibrium Transformer (GET) in offline distillation of diffusion models through a series of experiments on single-step class-conditional and unconditional image generation. Here, we use "single-step" to refer to the use of a single model evaluation while generating samples. We train and evaluate ViTs and GETs of varying scales on these tasks. GETs exhibit substantial data and parameter efficiency in offline distillation compared to the strong ViT baseline. Note that owing to the computational resources required to fully evaluate models, we report all our results on CIFAR-10 [52]; extensions to the ImageNet-scale [20] are possible, but would require substantially larger GPU resources.

## 4.1 Experiment setup

We will first outline our data collection process, followed by an in-depth discussion of our offline distillation procedure. We also include a brief summary of training details and evaluation metrics. For detailed network configs and training specifics, please refer to the Appendix.

**Data Collection.**    For unconditional image generation on CIFAR-10 [52], we generate 1M noise/image pairs from the pretrained unconditional EDM Karras et al. [48]. This dataset is denoted as `EDM-Uncond-1M`. As in EDM, we sample 1M images using Heun's second-order deterministic solver [3]. Generating a batch of images takes 18 steps or 35 NFEs (Number of Function Evaluations). Overall, this dataset takes up around 29 GB of disk space. The entire process of data generation takes about *4 hours* on 4 NVIDIA A6000 GPUs using Pytorch [74] Distributed Data Parallel (DDP) and a batch size of 128 per GPU. In addition to unconditional image generation, we sample 1M noise-label/image pairs from the conditional VP-EDM Karras et al. [48] using the same settings. This dataset is denoted as `EDM-Cond-1M`. Both the datasets will be released for future studies.

**Offline Distillation.**    We distill a pretrained EDM [48] into ViTs and GETs by training on a dataset $\mathcal{D}$ with noise/image pairs sampled from the teacher diffusion model using a reconstruction loss:

$$\mathcal{L}(\theta) = \mathbb{E}_{\mathbf{e},\mathbf{x}\sim\mathcal{D}}\|\mathbf{x} - G_\theta(\mathbf{e})\|_1$$

where $\mathbf{x}$ is the desired ground truth image, $G_\theta(\cdot)$ is unconditional ViT/GET with parameters $\theta$, and $\mathbf{e}$ is the initial Gaussian noise. To train a class-conditional GET, we also use class labels $\mathbf{y}$ in addition to noise/image pairs:

$$\mathcal{L}(\theta) = \mathbb{E}_{\mathbf{e},\mathbf{y},\mathbf{x}\sim\mathcal{D}}\|\mathbf{x} - G_\theta^c(\mathbf{e},\mathbf{y})\|_1$$

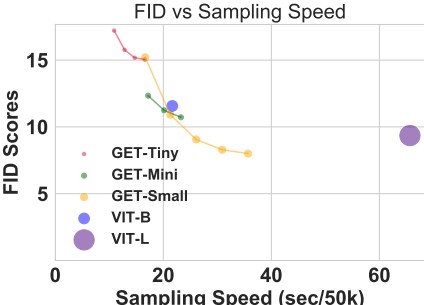
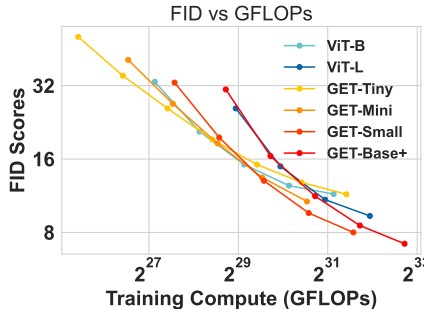

Figure 3: (a) (*Left*) **Sampling speed of GET**: GET can sample faster than large ViTs, while achieving better FID scores. The size of each individual circle is proportional to the model size. For GETs, we vary the number of iterations in the Equilibrium transformer (2 to 6 iterations). The trends indicate that GETs can improve their FID scores by using more compute. (b) (*Right*) **Compute efficiency of GET**: Larger GET models use training compute more efficiently. For a given GET, the training budget is calculated from training iterations. Refer to Table 2 for the exact size of GET models.

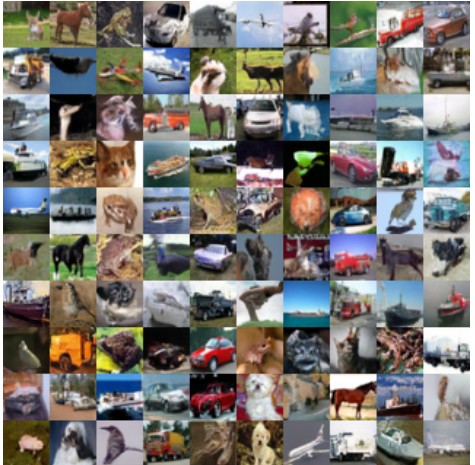
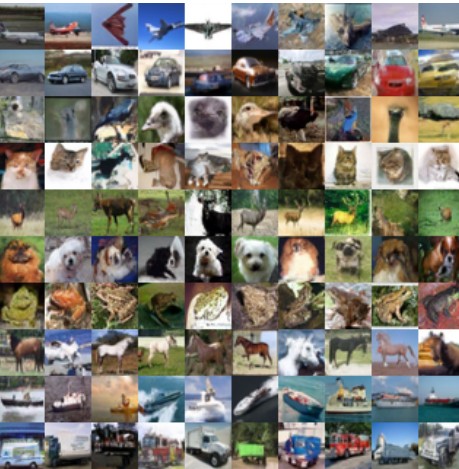

Figure 4: Uncurated CIFAR-10 image samples generated by *(Left)* (a) unconditional GET and *(Right)* (b) class-conditional GET. Each row corresponds to a class in CIFAR-10.

where $G_\theta^c(\cdot)$ is class-conditional ViT/GET with parameters $\theta$. As is the standard practice, we also maintain an exponential moving average (EMA) of weights of the model, which in turn is used at inference time for sampling.

**Training Details and Evaluation Metrics.**  We use AdamW [63] optimizer with a learning rate of 1e-4, a batch size of 128 (denoted as 1×BS), and 800k training iterations, which are identical to Progressive Distillation (PD) [88]. For conditional models, we adopt a batch size of 256 (2×BS). No warm-up, weight decay, or learning rate decay is applied. We convert input noise to patches of size $2 \times 2$. We use 6 steps of fixed point iterations in the forward pass of GET-DEQ and differentiate through it. For the $\mathcal{O}(1)$ memory mode, we utilize gradient checkpoint [17] for DEQ's computational graph. We measure image sample quality for all our experiments via Frechet inception distance (FID) [34] of 50k samples. We also report Inception Score (IS) [89] computed on 50k images. We include other relevant metrics such as FLOPs, training speed, memory, sampling speed, and the Number of Function Evaluations (NFEs), wherever necessary.

## 4.2   Experiment Results

We aim to answer the following questions through extensive experiments: 1) Can offline distillation match online distillation for diffusion models using GETs? 2) What is the scaling behavior of GET

as the model size and training compute increase? 3) How does GET compare to existing one-step generative models in terms of image quality and diversity?

**Efficiency.** Models trained with offline distillation require high data efficiency to make optimal use of limited training data sampled from pretrained diffusion models. DEQs have a natural regularization mechanism due to weight-tying, which allows us to efficiently fit significantly compact data-efficient models even in limited data settings. In Figure 2(a), we observe that even with a fixed and limited offline data budget of 1M samples, GET achieves parity with online distilled EDM [48, 88, 97] while using only half the number of training iterations. For comparison, PD, TRACT, and CM use a much larger data budget of 96M, 256M, and 409.6M samples, respectively. Moreover, GET is able to match the FID score of a 5× large ViT, suggesting substantial parameter efficiency.

**Sampling Speed.** Figure 3(a) illustrates the sampling speed of both ViT and GET. A smaller GET (37.2M) can achieve faster sampling than a larger ViT (302.6M) while achieving lower FID scores. GET can also improve its FID score by increasing its test-time iterations in the Equilibrium transformer at the cost of speed. Note that despite this trade-off, GET still outperforms larger VIT in terms of both sampling speed and sample quality.

**Why Scaling Laws for Implicit Models?** As a prospective study, we preliminarily investigate the scaling properties of Deep Equilibrium models using GET. The scaling law is an attractive property, as it enables us to predict models' performance at extremely large compute based on the performance of tiny models. This predictive capability allows us to select the most efficient model given the constraints of available training budget [13, 37, 72]. While the scaling law for explicit networks has been extensively studied, its counterpart for implicit models remains largely unexplored. Implicit models are different from explicit models as they utilize more computation through weight-tying under similar parameters and model designs. Therefore, it is natural to question whether their scaling laws align with those of their explicit counterparts.

**Scaling Model Size.** We conduct extensive experiments to understand the trends of sample quality as we scale the model size of GET. Table 2 provides a summary of our findings on single-step unconditional image generation. We find that even small GET models with 10-20M parameters can generate images with sample quality on par with NAS-derived AutoGAN [31]. In general, sample quality improves with the increase in model size.

**Scaling Training Compute.** Our experimental results support the findings of Peebles and Xie [75] for explicit models (DiT) and extend them to implicit models. Specifically, for both implicit and explicit models, larger models are better at exploiting training FLOPs. Figure 3 shows that larger models eventually outperform smaller models when the training compute increases. For implicit models, there also exists a "sweet spot" in terms of model size under a fixed training budget, *e.g.,* GET-Small outperforms both smaller and larger GETs at $2^{31}$ training GFLOPs. Furthermore, because of the internal dynamics of implicit models, they can match a much larger explicit model in terms of performance while using fewer parameters. This underscores the potential of implicit models as candidates for compute-optimal models [37] with substantially better parameter efficiency. For example, at $2^{31}$ training GFLOPs, Figure 3(b) suggests that we should choose GET-Small (31.2M) among implicit models for the best performance, which is much more parameter efficient and faster in sampling than the best-performing explicit model, ViT-L (302M), at this training budget.

**Benchmarking GET against ViT.** Table 4 summarizes key metrics for unconditional image generation for ViT and GET. Our experiments indicate that a smaller GET (19.2M) can generate higher-quality images faster than a much larger ViT (85.2M) while utilizing less training memory and fewer FLOPs. GET also demonstrates substantial parameter efficiency over ViTs as shown in Figure 2(b) where smaller GETs achieve better FID scores than larger ViTs.

**Comparizon of NFEs of teacher model.** Offline distillation requires significantly fewer number of function evaluations (NFEs) for the teacher network compared to other online distillation methods. In the experimental setup used in this paper, GET requires 35M overall NFEs for the teacher model, as we train on 1M data samples, and use 35 NFEs to generate each data sample with EDM. In contrast, progressive distillation requires 179M NFEs to get 1-step distilled student model. Using the hyperparameters reported in Salimans and Ho [88], PD with DDIM model needs 13 passes of

Table 1: Generative performance on unconditional CIFAR-10.

| Method | NFE ↓ | FID ↓ | IS ↑ |
|---|---|---|---|
| *Diffusion Models* | | | |
| DDPM [35] | 1000 | 3.17 | 9.46 |
| Score SDE [96] | 2000 | 2.2 | 9.89 |
| DDIM [94] | 10 | 13.36 | - |
| EDM [48] | 35 | 2.04 | 9.84 |
| *GANs* | | | |
| StyleGAN2 [47] | 1 | 8.32 | 9.18 |
| StyleGAN-XL [91] | 1 | 1.85 | - |
| *Diffusion Distillation* | | | |
| KD [66] | 1 | 9.36 | 8.36 |
| PD [88] | 1 | 9.12 | - |
| DFNO [111] | 1 | 4.12 | - |
| TRACT-EDM [11] | 1 | 4.17 | - |
| PD-EDM [88, 97] | 1 | 8.34 | 8.69 |
| CD-EDM (LPIPS) [97] | 1 | 3.55 | 9.48 |
| *Consistency Models* | | | |
| CT [97] | 1 | 8.70 | 8.49 |
| CT [97] | 2 | 5.83 | 8.85 |
| *Ours* | | | |
| GET-Base | 1 | 6.91 | 9.16 |

Table 2: Generative performance of GETs on unconditional CIFAR-10.

| Models | Params | NFE ↓ | FID ↓ | IS ↑ |
|---|---|---|---|---|
| GET-Tiny | 8.6M | 1 | 15.19 | 8.37 |
| GET-Mini | 19.2M | 1 | 10.72 | 8.69 |
| GET-Small | 37.2M | 1 | 8.00 | 9.03 |
| GET-Base | 62.2M | 1 | 7.42 | 9.16 |
| GET-Base+ | 83.5M | 1 | 7.19 | 9.09 |
| *More Training* | | | | |
| GET-Tiny-4×Iters | 8.9M | 1 | 11.47 | 8.64 |
| GET-Base-2×BS | 62.2M | 1 | 6.91 | 9.16 |

Table 3: Generative performance on class-conditional CIFAR-10. $w$ indicates the level of classifier guidance.

| Method | NFE ↓ | FID ↓ | IS ↑ |
|---|---|---|---|
| *GANs* | | | |
| BigGAN [12] | 1 | 14.73 | 9.22 |
| StyleGAN2-ADA [46] | 1 | 2.42 | 10.14 |
| *Diffusion Distillation* | | | |
| Guided Distillation ($w = 0$) [68] | 1 | 8.34 | 8.63 |
| Guided Distillation ($w = 0.3$) [68] | 1 | 7.34 | 8.90 |
| Guided Distillation ($w = 1$) [68] | 1 | 8.62 | 9.21 |
| Guided Distillation ($w = 2$) [68] | 1 | 13.23 | 9.23 |
| *Ours* | | | |
| GET-Base | 1 | 6.25 | 9.40 |

Table 4: Benchmarking GET against ViT on unconditional image generation on CIFAR-10. For the first time, implicit models for generative tasks *strictly* surpass explicit models in all metrics. Results are benchmarked on 4 A6000 GPUs using a batch size of 128, 800k iterations, and PyTorch [74] distributed training protocol. Training Mem stands for training memory consumed per GPU. $\mathcal{O}(1)$ symbolizes the $\mathcal{O}(1)$ training memory mode, which differs only in training memory and speed.

| Model | FID↓ | IS↑ | Params↓ | FLOPs↓ | Training Mem↓ | Training Speed↑ |
|---|---|---|---|---|---|---|
| ViT-Base | 11.49 | 8.61 | 85.2M | 23.0G | 10.1GB | 4.83 iter/sec |
| GET-Mini | 10.72 | 8.69 | 19.2M | 15.2G | 9.2GB | 5.79 iter/sec |
| GET-Mini-$\mathcal{O}(1)$ | - | - | - | - | 5.0GB | 4.53 iter/sec |

distillation. The initial 12 passes use 50K iterations, and the last pass uses 100K iterations. Each step of PD uses 2 teacher model NFEs. Thus, the overall number of teacher model NFEs can be evaluated as $2 \times 128$ (batch size) $\times$ (12 passes $\times$50K + 100K) = 179M samples. The number of NFEs of the teacher model increases to 1.433B if we assume that each of 8 TPUs use a batch size of 128. Consistency distillation [97] needs 409.6M teacher model NFEs (512 batch size $\times$ 800K iterations = 409.6M). In addition, the perceptual loss requires *double* NFEs as the teacher model.

**One-Step Image Generation.** We provide results for unconditional and class-conditional image generation on CIFAR-10 in Table 1 and Table 3, respectively. GET outperforms a much more complex distillation procedure—PD with classifier-free guidance—in class-conditional image generation. GET also outperforms PD and KD in terms of FID score for unconditional image generation. This effectiveness is intriguing, given that our approach for offline distillation is relatively simpler when compared to other state-of-the-art distillation techniques. We have outlined key differences in the experimental setup between our approach and other distillation techniques in Table 5.

We also visualize random CIFAR-10 [52] samples generated by GET for both unconditional and class-conditional cases in Figure 4. GET can learn rich semantics and world knowledge from the dataset, as depicted in the images. For instance, GET has learned the symmetric layout of dog faces solely using reconstruction loss in the pixel space, as shown in Figure 4(b).

Table 5: Comparison of relevant training and hyperparameter settings for common distillation techniques. GET requires neither multiple training phases nor any trajectory information. We only count the number of models involved in the forward pass and exclude EMA in #Models. † indicates offline distillation techniques. ▲For CD, we count the VGG network used in the perceptual loss [109].

| Model | FID ↓ | IS ↑ | BS | Training Phases | #Models | Trajectory | Teacher |
|---|---|---|---|---|---|---|---|
| KD [66]† | 9.36 | - | 4× | 1 | 1 | ✗ | DDIM |
| PD [88] | 9.12 | - | 1× | $\log_2(T)$ | 2 | ✓ | DDIM |
| DFNO [111]† | 4.12 | - | 2× | 1 | 1 | ✓ | DDIM |
| TRACT [11] | 14.40 | - | 2× | 1 | 1 | ✓ | DDIM |
| TRACT [11] | 4.17 | - | 2× | 2 | 1 | ✓ | EDM |
| PD-EDM [88, 97] | 8.34 | 8.69 | 4× | $\log_2(T)$ | 2 | ✓ | EDM |
| CD▲ [97] | 3.55 | 9.48 | 4× | 1 | 3 | ✓ | EDM |
| Ours† | 7.42 | 9.16 | 1× | 1 | 1 | ✗ | EDM |
| Ours† | 6.91 | 9.16 | 2× | 1 | 1 | ✗ | EDM |
| Guided Distillation [68] | 7.34 | 8.90 | 4× | $\log_2(T) + 1$ | 3 | ✓ | DDIM |
| Ours† | 6.25 | 9.40 | 2× | 1 | 1 | ✗ | EDM |

# 5 Related Work

**Distillation techniques for diffusion models.** Knowledge distillation (KD) [66] proposed to distill a multi-step DDIM [94] sampler into the pretrained UNet by training the student model on synthetic image samples. There are several key differences from this work: Our approach does not rely on temporal embeddings or generative pretrained weights and predicts images instead of noises. Further, GET is built upon ViT [25], unlike the UNet in KD. Additionally, we demonstrate the effectiveness of our approach on both unconditional and class-conditional image generation.

Progressive distillation (PD) [88] proposes a strategy for online distillation to distill a $T$-step teacher DDIM [94] diffusion model into a new $T/2$ step student DDIM model, repeating this process until one-step models are achieved. Transitive closure time-distillation (TRACT) [11] generalizes PD to distill $N > 2$ steps together at once, reducing the overall number of training phases. Consistency models [97] achieve online distillation in a single pass by taking advantage of a carefully designed teacher and distillation loss objective.

Diffusion Fourier neural operator (DFNO) [111] maps the initial Gaussian distribution to the solution trajectory of the reverse diffusion process by inserting the temporal Fourier integral operators in the pretrained U-Net backbone. Meng et al. [68] propose a two-stage approach to distill classifier-free guided diffusion models into few-step generative models by first distilling a combined conditional and unconditional model, and then progressively distilling the resulting model for faster generation.

**Fast sampler for diffusion models.** While distillation is a predominant approach to speed up the sampling speed of existing diffusion models, there are alternate lines of work to reduce the length of sampling chains by considering alternate formulations of diffusion model [48, 50, 94, 96, 103], correcting bias and truncation errors in the denoising process [9, 10, 90], and through training-free fast samplers at inference [24, 43, 50, 59, 65, 108]. Several works like Improved DDPM [70], SGM-CLD [23], EDM [48] modify or optimize the forward diffusion process so that the reverse denoising process can be made more efficient. Diffusion Exponential Integrator Sampler (DEIS) [108] uses an exponential integrator over the Euler method to minimize discretization error while solving SDE. DPM-Solver [65], and GENIE [24] are higher-order ODE solvers that generate samples in a few steps.

# 6 Limitations

Our method for offline distillation relies on deterministic samplers to ensure a unique mapping between initial noise **e** and image **x**. As a result, it cannot be directly applied to stochastic samplers which do not satisfy this requirement. However, this limitation also applies to many other distillation techniques, as they cannot maintain their fidelity under stochastic trajectories [11, 66, 88, 97].

# 7 Conclusion

We propose a simple yet effective approach to distill diffusion models into generative models capable of sampling with just a single model evaluation. Our method involves training a Generative Equilibrium Transformer (GET) architecture directly on noise/image pairs generated from a pre-trained diffusion model, eliminating the need for trajectory information and temporal embedding. GET demonstrates superior performance over more complex online distillation techniques such as progressive distillation [68, 88] in both class-conditional and unconditional settings. In addition, a small GET can generate higher quality images than a $5\times$ larger ViT, sampling faster while using less training memory and fewer compute FLOPs, demonstrating its effectiveness.

# 8 Acknowledgements

Zhengyang Geng and Ashwini Pokle are supported by grants from the Bosch Center for Artificial Intelligence.

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

## A   Additional Experiments

**Class Conditioning.**    As both GET and ViT share the same class injection interface, we perform an ablation study on ViT. We consider two types of input injection schemes for class labels: 1) additive injection scheme 2) injection with adaptive layer normalization (AdaLN-Zero) as used in DiT [75]. Despite using almost the same parameters as unconditional ViT-B, the class-conditional ViT-B using additive injection interface has an FID of 12.43 at 200k, while the ViT-B w/ AdaLN-Zero class embedding [75] set up an FID of 17.19 at 200k iterations. Another surprising observation is that ViT-B w/ AdaLN-Zero class embedding performs worse than unconditional ViT in terms of FID score. Therefore, it seems that adaptive layer normalization might not be useful when used only with class embedding.

Table 6: Ablation on class conditioning.

| Model | FID↓ | IS↑ | Params↓ |
|---|---|---|---|
| ViT-Uncond | 15.20 | 8.27 | 85.2M |
| ViT-AdaLN-Zero | 17.19 | 8.38 | 128.9M |
| ViT-Inj-Interface | 12.43 | 8.69 | 85.2M |

## B   Model Configuration

We set the EMA momentum to $0.9999$ for all the models.

The configuration of different GET architectures are listed in Table 7. Here, $L_i$ and $L_e$ denote the number of transformer blocks in the Injection transformer and Equilibrium transformer, respectively. $D$ denotes the width of the network. $E$ corresponds to the expanding factor of the FFN layer in the Equilibrium transformer, which results in the hidden dimension of $E \times D$. For the injection transformer, we always adopt an expanding factor of 4.

Table 7: Details of configuration for GET architectures.

| Model | Params | $L_i$ | $L_e$ | $D$ | $E$ |
|---|---|---|---|---|---|
| GET-Tiny | 8.9M | 6 | 3 | 256 | 6 |
| GET-Mini | 19.2M | 6 | 3 | 384 | 6 |
| GET-Small | 37.2M | 6 | 3 | 512 | 6 |
| GET-Base | 62.2M | 1 | 3 | 768 | 12 |
| GET-Base+ | 83.5M | 6 | 3 | 768 | 8 |

We have listed relevant model configuration details of ViT in Table 8. The model configurations are adopted from DiT [75], whose effectiveness was tested for learning diffusion models. In this table, $L$ denotes the number of transformer blocks in ViT. $D$ stands for the width of the network. We always adopt an expanding factor of 4 following the common practice [25, 75, 99].

Table 8: Details of configuration for ViT architectures.

| Model | Params | $L$ | $D$ |
|---|---|---|---|
| ViT-B | 85.2M | 12 | 768 |
| ViT-L | 302.6M | 24 | 1024 |

## C   Related Work

**Transformers.**    Transformers were first proposed by Vaswani et al. [99] for machine translation and since then have been widely applied in many domains like natural language processing [21, 55, 79, 84], reinforcement learning [16, 73], self-supervised learning [15], vision [25, 62], and generative

modeling [26, 41, 75, 82]. Many design paradigms for transformer architectures have emerged over the years. Notable ones include encoder-only [21, 54, 61], decoder-only [13, 79, 80, 101, 104], and encoder-decoder architectures [53, 81, 99]. We are interested in scalable transformer architectures for generative modeling. Most relevant to this work are two encoder-only transformer architectures: Vision Transformer (ViT) [25] and Diffusion Transformer (DiT) [75]. Vision Transformer (ViT) closely follows the original transformer architecture. It first converts 2D images into patches that are flattened and projected into an embedding space. 2D Positional encoding is added to the patch embedding to retain positional information. This sequence of embedding vectors is fed into the standard transformer architecture. Diffusion Transformers (DiT) are based on ViT architecture and operate on sequences of patches of an image that are projected into a latent space through an image encoder [85]. In addition, DiTs adapt several architectural modifications that enable their use as a backbone for diffusion models and help them scale better with increasing model size, including adaptive Layer Normalization (AdaLN-Zero) [12, 22, 45, 76] for time and class embedding, and zero-initialization for the final convolution layer [32].

**Deep equilibrium models.** Deep Equilibrium models (DEQs) [5] solve for a fixed point in the forward pass. Specifically, given the input $\mathbf{x}$ and the equilibrium function $f_\theta$, DEQ models approach the infinite-depth representation of $f_\theta$ by solving for its fixed point $\mathbf{z}^\star$: $\mathbf{z}^\star = f_\theta(\mathbf{z}^\star, \mathbf{x})$. For the backward pass, one can differentiate analytically through $z^\star$ by the implicit function theorem. The training dynamics of DEQ models can be unstable for certain model designs [7]. As a result, recent efforts focus on addressing these issues by designing variants of DEQs with provable guarantees [83, 105], or through optimization techniques such as Jacobian regularization [7], and fixed-point correction [8]. DEQs have been successfully applied on a wide range of tasks such as image classification [6], semantic segmentation [6, 110], optical flow estimation [8], object detection [100, 102], landmark detection [69], out-of-distribution generalization [2], language modelling [5], input optimization [33], unsupervised learning [98], and generative models [64, 77].

