# OpenReview forum: "One-Step Diffusion Distillation via Deep Equilibrium Models"
_NeurIPS.cc/2023/Conference — NeurIPS 2023 poster_

### Official Review · Reviewer_MHac · 2023-06-29

**Soundness:** 3 good
**Presentation:** 3 good
**Contribution:** 2 fair
**Rating:** 5
**Confidence:** 4

**Summary:**

This paper applies Deep Equilibrium Models to distillation of (conditioned) diffusion models. The key part of the architecture consists of a repeated application of weight-tied block of layers on the internal activations (theoretically until convergence to a fixed point, but in practice a few iterations). The model is trained to mimic the mapping from noise to images induced by the diffusion sampling chain.

**Strengths:**

The premise of the paper makes sense. One important reason for the success of diffusion models might be that the chain of repeated (though time-varying) neural steps is, in aggregate, able to represent extremely complicated mappings, with near-discontinuous regions packed in complex high-dimensional shapes, etc. – something that is amply present in the latent-to-image mapping induced by the diffusion flow. Capturing this complexity with typical single-pass networks of modest layer counts is challenging, as evidenced by many attempts in the literature. DEQ seems like an appealing and natural approach here, as it has the similar character of iteratively shaping an initially simple distribution into a very complex one. There is also the promise of parameter count savings: effectively they implement a more complex function than just a single layer block, but parametrized on the same amount of weights.

The proposed network architectures seem well justified for this task.

The evaluation makes a reasonably convincing case that this can be beneficial, and that the promised benefits of DEQs are realized up to a point. There is a decent amount of useful experimentation and attempt to understand scaling around model size and other parameters.

**Weaknesses:**

The paper is scarce on details about the DEQ training and inference. The impression I’m left with (based on line 213 and the fact that no details are provided), is the authors do not actually use almost any of the DEQ machinery, such as fixed point solvers and related backpropagation shortcuts. Instead, the model seems to be trained as though it were a regular neural network, where the same block of layers is stacked in sequence six times (but with tied weights between them). This may work well in practice but raises some questions about whether DEQs are as relevant to the story as claimed, and whether some of their potential advantages are left on the table. There does seem to be fixed-point like behavior and capability though, as illustrated in Figure 3a.

Nevertheless, this raises the question about some baselines. What if the weights were _not_ tied? Wouldn’t the inference and training remain equal number of flops to what you currently do, but with some modest extra memory consumption for storing the weight values (but importantly, no more or less activations to store)? By default, wouldn’t one expect this to strictly extend the capability of the model? It’s possible that the tying has some implicit regularization effects like discussed on line 224, but this claim is speculative. It might also make the training more effective (as in, converge with fewer iterations), but this is also not clear-cut. The evaluation does not currently explore or rule out the possibility that weight tying could even be harming an otherwise higher performing model. Such a baseline would distinguish whether the benefit comes from the particular architecture and depth, or from the DEQ-like aspects. At present, I am somewhat uncertain about which it is – the current ViT baseline seems sufficiently different that it might underperform for unrelated reasons.

This would of course increase the parameter count, but note that parameter count is often taken as a proxy for evaluation time and memory consumption, neither of which would be seriously affected here. That is, unless my understanding is wrong, and the paper actually does use DEQ training mechanisms. In this case, it is missing a significant amount of important detail about how those are implemented and how the associated difficulties are dealt with (this should be clarified in either case).

The water is also a bit muddy in terms of how many iterations any given distilled model requires. One could argue that this is in spirit a six-step model (flanked by pre and post-processing to embed in a latent space), or conversely, that e.g. a four-step progressive distillation model is actually just a one-step model that internally calls some layers four times. The more relevant question comes down to flops and wall-clock time rather than the semantics of what we take the iterations to mean. Accordingly, including something like 2 or 4 step PD (which beat the metrics here) in the comparisons would be reasonable, unless it can be convincingly argued that their runtime is significantly slower due to the sub-stepping (I am unsure of whether this is the case).

**Questions:**

Beyond the baselines discussed above, the role of the fixed point remains somewhat unclear either way. Does the iteration converge to a fixed point in six iterations? Could it even diverge? If one were to evaluate it all the way to a fixed point, how much extra performance could be squeezed out? Is six steps enough in training to reap these benefits?

Miscellaneous observation: are you sure you are citing the correct FID/IS for Progressive Distillation [61] in unconditional CIFAR (Table 1)? If I understand correctly, the number you cite is the conditional model (Table 4 in appendix of [50]), which benefits from the conditioning info despite lack of guidance boost. However, for the unconditional model, Table 2 in [61] cites a FID of 9.12. Wouldn’t this be the correct number (not sure)? Related, I’m not sure why this model is referred to as PD-EDM, and cited in combination with Consistency Models?

**Limitations:**

There is a brief discussion of limitations, but it does not go into much depth.

---

> ### Author Rebuttal · Authors · 2023-08-10
>
> Thank you for your extremely thoughtful feedback and suggestions. We have tried our best to answer your questions and concerns below.
>
> > Does the iteration converge to a fixed point in six iterations?
>
> We report the relative fixed point error $\frac{\|| f(z) - z \||} {\|| f(z) \||}$ and the resulting FID score as we vary the number of iterations of the equilibrium transformer within GET-M at inference time.
>
> | Iterations | Relative Error  |  FID |
> |----------|:-------------:|------:|
> | 1 |  1.028 |  42.96 |
> | 2 |  0.334 |  26.70 |
> | 3 |  0.215 |  20.27 |
> | 4 |  0.148 |  17.21 |
> | 5 |  0.107 |  15.77 |
> | 6 |  0.082 |  15.18 |
> | 7 |  0.066 |  15.04 |
> | 8 |  0.055 |  15.18 |
> | 9 |  0.048 |  15.46 |
> | 10 |  0.043 | 15.85 |
>
> Based on our empirical results, we have indeed converged to a fixed point with a relatively low relative fixed point error of 0.082 within 6 iterations. As a comparison, DEQ-transformer [1] trained with gradients computed via implicit function theorem requires 30 iterations to achieve a relative error of 0.10 on language modeling.
>
> We also observe improved FID scores as we increase the depth to reach the depth at training time. The FID score eventually flattens out and marginally increases as we further increase the depth. As the reviewer has noted, it is certainly possible to get improved performance from GET by marginally increasing the depth at inference time. For instance, GET-M’s FID score improves to 15.04 when we increase the iterations at inference time to 7.
>
> __Does weight-tying harm GET’s performance?__:
>
> The reviewer has valid concerns on whether weight tying harms GET’s performance. To address this concern, we trained and evaluated a non-weight tied architecture. In this architecture, we unroll the equilibrium transformer in GET, effectively using six separate GET blocks, instead of iterating over a single GET block within the equilibrium transformer. This architecture is otherwise exactly similar to GET’s architecture, and thus has the same number of FLOPs. We trained this model for 800k iterations, similar to the training duration of the weight-tied model, and found this model was 16%  slower in training, despite having the same FLOPs. Further, This model also achieves an inferior FID score and Inception score compared to the weight-tied model. This outcome underscores that weight-tying within GET indeed contributes to improved performance.
>
> | Model | Params |  FID  |  IS  |
> |----------|:-------------:|:----:| :----:|
> | GET-B   	    | 62.2M | 7.42 | 9.16 |
> | GET-B-non-WT | 310M  | 7.55 | 9.08 |
>
> __Comparison of runtime of GET against PD__:
>
> To ensure a fair comparison between GET and PD, we need to benchmark GET against PD baseline that uses ViT/DiT architecture. However, we do not have access to a ViT/DiT model that is distilled via PD on CIFAR-10. As a proxy for PD, we report the sampling time of EDM (50k images, 1NFE). A single forward pass through EDM requires 27s, whereas GET-Tiny completes sampling in 14.7s, GET-Mini takes 23.3s, and GET-Small needs 35.7s (all under 6 iterations).
>
> Considering that we use significantly fewer teacher model evaluations than PD (35M vs 179M (lower bound) or 1.433B (upper bound)), the performance can potentially be improved by using more training data and by increasing training iterations. As an example, the GET-Tiny with 8.9M parameters and 14.7s sampling time can reach FID of 11.47 and IS of 8.64 (while PD-EDM has IS of 8.69 at > 410M teacher evaluations), given 4$\times$ training iterations on *the same data* (35M teacher evaluations).
>
> [1] Bai, Shaojie, J. Zico Kolter, and Vladlen Koltun. "Deep equilibrium models." Advances in Neural Information Processing Systems 32 (2019).
>
> > Are you sure you are citing the correct FID/IS for Progressive Distillation [61] in unconditional CIFAR (Table 1)?
>
> Thank you for requesting this clarification. In Table 1, we have cited PD [61] at FID 9.12 (second row in Diffusion Distillation). We have also cited guided distillation [50] in Table 2, which is the performance for class-conditional models. The results for PD-EDM in Table 1 have been reported by Consistency Models [70] using EDM and stronger training settings, and we include it as a relevant baseline.
>
> We are happy to answer any further questions or concerns that you may have.

---

> > ### Comment · Reviewer_MHac · 2023-08-18
> > **Rebuttal response**
> >
> > Thank you for the thoughtful responses and experiments -- these results address my main concerns about potential hidden issues. Incorporating these and other clarifications in the paper and/or appendix will improve the paper. I remain leaning somewhat positive on acceptance.

---

### Official Review · Reviewer_sYNQ · 2023-07-06

**Soundness:** 2 fair
**Presentation:** 3 good
**Contribution:** 2 fair
**Rating:** 5
**Confidence:** 3

**Summary:**

The paper proposes a simple approach to distill diffusion models into generative models capable of sampling with just a single model evaluation. The method involves training a Generative Equilibrium Transformer (GET) architecture directly on noise/image pairs generated from a pre-trained diffusion model, eliminating the need for trajectory information and temporal embedding. The experiments verify its effectiveness.

**Strengths:**

- The paper is well-written and easy to follow.
- The idea of using DEQ for distillation is interesting.

**Weaknesses:**

To verify the effectiveness, the results of ImageNet are neccessary, since ImageNet is another mainstream benchmark. Some experimental design are not convincing enough. More details please refer to the questions.

**Questions:**

- Could you explain more about the motivation and advantages of using DEQ?
- Following the prior question, about the comparison in Table 4: have you tried the diffusion transformer (DiT), which may be more parameter-efficient for denoising process? The same question for the scaling experiments.
- In Figure 2(Left), the performance of GET falls off quickly with less iterations in the Equilibrium transformer. Compared to other distillation methods such as PD and CD with single step, what are the advantages? Are there any results about the sampling speed using different iterations between PD(CD) and GET?
- Instead of 6 iterations, have you tried two GET models for two stage of denoising steps and used 2 iterations for two GET models sperately?($t$ from 0 to 0.5 and $t$ from 0.5 to 1.0). This may be more efficient than using 6 iterations.
- Does GET drop condition during training and support classifier-free guidance? Can this technique improve the performance?

**Limitations:**

The authors have adequately addressed the limitations.

---

> ### Author Rebuttal · Authors · 2023-08-10
>
> We thank the reviewer for well-thought questions and valuable feedback. We have tried our best to answer your questions.
>
> __Motivation and Advantages of DEQ__:
> Our motivation to model the student network as a DEQ stems from the observation that the relatively complex process of distilling diffusion models has an element of fixed point process, as indicated by recent works like Consistency Models (Song et al. 2023) and TRACT (Berthelot et al. 2023).  In this work, we sought a network architecture capable of adapting its compute requirements according to the complexity of image being generated. DEQs possess both of these capabilities. The empirical results in our paper indeed indicate that modeling the student network (i.e. GET) as a DEQ is vital to achieve good performance in our set-up of one-step distillation.
>
> __Benchmarking against DiT__: In Table. 4, the ViT baseline was simplified from the diffusion transformer (DiT). Because our distillation strategy does not require the temporal embedding, we remove this layer from DiT, leading to the ViT baseline shown in Table. 4.
>
> __Clarification about one-step generation__: We define one-step generation as the ability to generate an image directly from Gaussian noise in a _single_ forward pass through the network.  There can be multiple iterations over the equilibrium transformer during this forward pass but it is important to note that all of these iterations contribute to the depth of the network. In Figure 3(a), we demonstrate how the performance changes with variation in the effective depth of GET. Standard neural network architectures (e.g., ResNet, Diffusion Transformer (DiT)) increase the depth of the network by applying a block multiple times, increasing the number of parameters in the process. In contrast, GET weight-ties the repeated block, preserving the overall number of parameters.
>
> __Variation of GET’s performance with iterations__: As shown in Figure 3a (left), GET’s performance improves with more iterations because of the increase in effective depth of the network. However, this computation still constitutes a single forward pass through the network, and thus one-step generation.
>
> __Advantages of GET over other distillation methods such as PD and CD with single step__:
>
> __Advantages over PD__: GET has several advantages over Progressive Distillation (PD).
> - _Complexity_: On CIFAR10, PD uses 13 training passes to distill 8192-step DDIM chains to 1 step. In addition, PD also carefully selects distillation targets in each step.
> - _Performance_: One-step GET (that uses a single forward pass) outperforms PD for both class-conditional and unconditional cases on CIFAR-10 in terms of FID score and Inception Score.
> - _Efficiency_: On CIFAR10, PD consumed 179M teacher model evaluations (2 DDIM steps * 128 (batch size) * (12 passes * 50k + 100K) = 179M), while GET only consumed 1M samples * 35 NFE = 35M teacher model evaluations.
>
> __Advantages over CD__: CD uses far more complex distillation settings compared to GET. As shown in our paper,
> - _Complexity_: CD leverages trajectory information from the teacher diffusion model and employs perceptual loss (LPIPS), which results in forward passes through 3 models during the distillation process.
> - _Efficiency_: On CIFAR10, CD requires significantly more teacher models evaluations  (512 (batch size) * 800k iterations = 409.6M evaluations) compared to our method which consumes 1M samples * 35 NFE = 35M teacher model evaluations.
> - _Data requirement_: CD assumes access to the dataset on which the original diffusion model was trained. In contrast, GET can be trained fully on synthetic data. This is beneficial in cases where we do not have access to the original dataset on which the diffusion model was trained.
> - _Auxiliary loss_: CD uses LPIPS perceptual loss as an auxiliary loss during training. In the absence of this loss, CD’s performance drops considerably (even doubles). Please see Figure 4 in CD’s paper.
>
> __Sampling speed of GET against PD and CD__:
> To ensure a fair comparison between GET and PD, we need to benchmark GET against PD baseline that uses ViT/DiT architecture. However, we do not have access to a ViT/DiT model that is distilled via PD on CIFAR-10. Similar concerns hold true for CD. As a proxy, we report the sampling time of EDM (50k images, 1 NFE). A single forward pass through EDM requires 27s, whereas GET-Tiny completes sampling in 14.7s, GET-Mini takes 23.3s, and GET-Small needs 35.7s (all under 6 iterations).
>
> Considering that we use significantly fewer teacher model evaluations than PD (35M vs 179M (lower bound) or 1.433B (upper bound)), the performance can potentially be improved by using more training data and by increasing training iterations. As an example, the GET-Tiny with 8.9M parameters and 14.7s sampling time can reach FID of 11.47 and IS of 8.64 (while PD-EDM has IS of 8.69 at > 410M teacher evaluations), given 4$\times$ training iterations on *the same data* (35M teacher evaluations).
> In case, there is a publicly available model that the reviewer would like us to benchmark against, we are happy to evaluate it.
>
> __Two stage denoising with GET__: GET learns a direct mapping between noise and image pairs without any use of temporal embeddings. As a result, we cannot stack two GET models to get two-state denoising. Further, in our paper, we demonstrate empirically that GET benefits from its DEQ architecture where we solve for a fixed point. Reducing the number of iterations in the Equilibrium transformer to 3 will lead to poor convergence of this fixed point, and might adversely affect the performance of GET.
>
> __Classifier-free guidance while training GET__: Thanks for this suggestion! It is certainly possible to train GET with classifier guidance. This is an interesting experiment that we will try in the future.

---

> > ### Comment · Reviewer_sYNQ · 2023-08-21
> > **Thanks for your responses.**
> >
> > Thanks for the responses and experiments. The overall idea is quite interesting and I think my questions have been well addressed. Therefore, I am willing to increase my score to positive side.

---

### Official Review · Reviewer_FiXo · 2023-07-08

**Soundness:** 2 fair
**Presentation:** 3 good
**Contribution:** 3 good
**Rating:** 6
**Confidence:** 4

**Summary:**

The submission proposes the Generative Equilibrium Transformer (GET), a lightweight refinement of vision transformer that is well-suited as an efficient single-step student model for diffusion distillation. The author empirically shows that the GET outperforms classic networks in terms of performance, model size and inference efficiency. Overall, the introduced architecture improvements shed light on further studies for improving student models of diffusion distillation.

**Strengths:**

1. The introduction of DEQ for efficient students is novel: The idea of introducing the DEQ model into the design of a more efficient student model is novel. Till now, most diffusion distillation works take the same architecture of the student model as the teacher model. Few of them have investigated the design of an efficient student model, which is potentially important for applications with strict requirements on inference efficiency. The author brings the DEQ model to design a ViT-based encoder-decoder student network and shows promising performance on CIFAR10 and ImageNet64 data.
2. Good Writing: The writing of this paper is clear and easy to follow.

**Weaknesses:**

Insufficient evaluations: the paper proposes an architecture-level improvement, the GET, that is related to ViT. The readers will be more convinced that the GET should be compared with the ViT-based diffusion model on modeling tasks on a large scale, such as the DiT on ImageNet128 and 256. I think the pros and cons of the proposed GET can be observed more clearly for large-scale benchmarks. Besides, the one-step student model shows an FID of 6.91 on the CIFAR10 dataset, which is significantly worse than Consistency Distillation which has an FID of 3.55. I am curious about the possibility to combine GET with other distillation methods such as CD to obtain stronger performance.

**Questions:**

See the weakness above.

**Limitations:**

The author has discussed the limitations.

---

> ### Author Rebuttal · Authors · 2023-08-10
>
> Thank you for your encouraging feedback! Please find our responses to your questions and concerns below.
>
> __Extensive Large Scale Evaluations__: It is certainly possible to scale up to larger datasets like ImageNet but this would require significantly more computing resources. For example, even training Progressive Distillation on ImageNet 64 would consume 5 days on 64 TPUv4 chips, which is equivalent to/more than 7680 A100 hours. We have tried our best to demonstrate the scaling capabilities of GET in terms of training FLOPs and parameters (See Figure 3a/b) to support the possibility of transferring GET to more complex settings.
>
> __Comparison with CD__: Thanks for pointing out that CD is a strong baseline. However, CD uses far more complex distillation settings compared to GET. As shown in our paper,
> - _Complexity_: CD leverages trajectory information from the teacher diffusion model and employs perceptual loss (LPIPS), which results in forward passes through 3 models during the distillation process.
> - _Efficiency_: On CIFAR10, CD requires significantly more teacher models evaluations  (512 (batch size) * 800k iterations = 409.6M evaluations) compared to our method which consumes 1M samples * 35 NFE = 35M teacher model evaluations.
> - _Data requirement_: CD assumes access to the dataset on which the original diffusion model was trained. In contrast, GET can be trained fully on synthetic data. This is beneficial in cases where we do not have access to the original dataset on which the diffusion model was trained.
> - _Auxiliary loss_: CD uses LPIPS perceptual loss as an auxiliary loss during training. In the absence of this loss, CD’s performance drops considerably (even doubles). Please see Figure 4 in CD’s paper.
>
> __Combining GET with other distillation methods__: We agree that combining GET with other distillation techniques like consistency distillation is a promising direction. However, pursuing this avenue necessitates training GET as a diffusion model with denoising loss, and followed by distilling it into a few-step generative model using CD or PD. This process will require multiple model training phases, rendering the overall process more complex. This complexity runs  contrary to our initial motivation of using a DEQ-based architecture for diffusion distillation in a single-forward pass.
>
> We are happy to answer any follow up questions that you might have.

---

> > ### Comment · Reviewer_FiXo · 2023-08-19
> >
> > Thank you for authors’ reply. I understand that large-scale experiment may be a too strict experiment in the rebuttal period. I think the submission has demonstrated clearly its proposed GET architecture and made enough explorations on each component of GET on CIFAR10 dataset. Overall, their GET have shown promising performances with significantly fewer parameters and computational costs (both for training and inferences) than default UNet architectures. I think the work of GET may have high compact on elucidating the space of student models for diffusion distillation, so the work has significant novelty.
> >
> > Combining with its neat writing and clear motivation, I decide to raise my score to 6 to recommend the submission for acceptance.
> >
> > I think it would be nice (not necessary), if the author could use their proposed GET for training diffusion models (potentially with additional time-embedding layers), instead of distilling existing diffusion models, to explore the possibility of the GET architecture on more general applications.

---

### Official Review · Reviewer_GPxt · 2023-07-08

**Soundness:** 3 good
**Presentation:** 4 excellent
**Contribution:** 4 excellent
**Rating:** 5
**Confidence:** 3

**Summary:**

This paper proposes a new model, called Generative Equilibrium Transformer (GET). GET is a deep equilibrium model, trained as an implicit model, to match noise/image pairs generated with a pretrained diffusion model, and thereby distill that pretrained (multi-step) diffusion model into a fast, single-step approach. Experiments on CIFAR-10 demonstrate that this approach is able to distill diffusion models into a fast, efficient architecture that strictly outperforms an explicit implementation.

**Strengths:**

This is a nice paper, with promising results on distillation of diffusion models into very fast, efficient models. Although only conducted on a relatively small-scale dataset, experiments hint at scalability, efficiency; strictly outperforming an explicit counterpart (implemented as a ViT).

**Weaknesses:**

Overall, this is a strong paper with thorough experiments. A few points, however, remain unaddressed:
- How does the approach depend on the number of noise/data samples synthesized with the pretrained diffusion model? For conditional models, do we need to adapt the sampling strategy?
- While experiments on scalability of the model size are conducted on a small scale, the paper would benefit massively from applying the method to some larger pretrained model, ideally on a more demanding task like text-to-image (although intermediate steps on either (i) higher-resolution images, e.g. LSUN, or FFHQ, or more complex tasks like class-conditional ImageNet could suffice).

For further comments, please refer to the "Questions" section.

**Questions:**

- Can the DEQ approach be used to distill a pretrained UNet (like Stable Diffusion) into a DET?
- Why no experiments on convolutional architectures, which are very widely used in the community? For example, could a DEQ be initilaized from a pretrained Unet?

**Limitations:**

adressed.

---

> ### Author Rebuttal · Authors · 2023-08-10
>
> We are encouraged to know that the reviewer feels that this paper is strong with thorough experiments! We have tried our best to answer your questions below.
>
> __Variation of performance with number of samples__:
> Given this is a supervised learning set up, we anticipate that GET’s generalization will improve as we increase the number of noise/image pairs, up to a certain threshold, beyond which saturation is likely to occur.  To test this hypothesis empirically, we sampled an additional 1M data pairs for GET in the conditional setting and trained GET-B using the total 2M data. The resulting model achieved an FID of **5.66** and IS of **9.63**, while the model trained on 1M data has an FID of 6.23 and IS of 9.42. We believe that further scaling up of training data is certainly possible. GET might also potentially benefit from better supervision through perceptual loss.
>
> __Sampling strategy for conditional models__: For class-conditional sampling, we need to provide the class label in addition to the initial Gaussian noise as inputs to GET. GET generates images in a single forward pass for both class-conditional and unconditional cases. As GET learns a direct mapping from the initial noise and conditioning to the resulting images, we don’t need to adapt a special sampling strategy to generate class-conditioned images. This is a potential advantage of GET over distillation methods like PD in conditional settings.
>
> __Large scale scalability experiments__: We agree that conducting experiments on larger datasets would indeed provide more comprehensive insights into the scaling capabilities of GET. However, we lack computational resources to perform such large scale experiments. For instance, consistency distillation uses 64 Nvidia A100 GPUs for experiments on ImageNet-64x64 and LSUN 256x256. Progressive distillation reports use of 64 TPUv4 chips for their large scale experiments on ImageNet-64X64 and LSUN 256X256.
>
> __Can the DEQ approach be used to distill a pretrained UNet (like Stable Diffusion) into a GET?__ We can distill any pretrained diffusion model regardless of the network architecture into GET as long as we have noise/image pairs from the pretrained diffusion model.
>
> __Experiments on convolutional architectures like UNet__: Offline distillation for convolutional architectures has been explored by previous works like knowledge distillation (Luhman & Luhman, 2021). Knowledge distillation distills DDIM sampling chains with 1000 steps and uses UNet architecture. We outperform knowledge distillation on metrics like FID Score and Inception Score.
>
> __Initializing GET with pretrained UNet__: While we cannot initialize GET network with weights of a pretrained UNet, we can use noise/image pairs from (multi-step) UNet to distill diffusion trajectories into GET.
>
> Please let us know if you have any follow up questions and other concerns. We are happy to answer them!

---

> > ### Comment · Reviewer_GPxt · 2023-08-19
> >
> > Thanks for your response!
> > Let me clarify my question on "Initializing GET with pretrained UNet". What I meant is, why not apply DEQ-style training to a convolutional architecture? This should be possible in my understanding but happy to be corrected here.
> >
> > I still think that the paper would benefit very much from being applied to either (i) distillation of a strong, pretrained txt2img model or (ii), and as mentioned by other reviewers, to high-res datasets.

---

> > > ### Author Response · Authors · 2023-08-19
> > > **Reply to Reviewer GPxt**
> > >
> > > Thank you for your insightful suggestions and for your clarification on "initializing GET with pretrained UNet"! We apologize for the misunderstanding regarding your question. We sincerely hope to address your question regarding the convolutional scheme.
> > >
> > > We agree that DEQ-style architecture training can indeed be extended to convolutional designs. It is a bit non-trivial to convert UNet into a DEQ, as it has cross-connections across upsampling and downsampling blocks, which also contributes to its efficiency for image generation tasks.
> > >
> > > While we did experiment with the MDEQ [1] architecture, a purely convolutional design, this architecture did not perform on par with UNet/ViT for image generation. The UNet structure, particularly within the realm of diffusion models, has been iteratively refined by researchers over time. Converting a UNet into the DEQ framework would require designing a proper input injection scheme (as it has multiple downsampling and upsampling stages and cross-stage connections) and improving its gradient flow in the backward pass.
> > >
> > > Thus, adapting the DEQ to convolutional designs remains a promising and challenging topic for future research. Further explorations are required to pinpoint the optimal architectural elements like the injection layer that could potentially outperform, or at least match, the convolutional UNet. In this work, we thus decided to focus on GET and demonstrate its benefits in the context of distilling diffusion models.
> > >
> > > [1] Bai, Shaojie, Vladlen Koltun, and J. Zico Kolter. "Multiscale deep equilibrium models." Advances in Neural Information Processing Systems 33 (2020): 5238-5250.
> > >
> > > We appreciate your suggestions on text2img generation and experiments with high-resolution datasets!
> > > However, these experiments demand substantial computational resources beyond our current reach.
> > > For example, even with ImageNet 64x64 images, progressive distillation requires 5 days of training across 64 TPUv4 chips, approximately 7680 TPU hours. Considering a similar amount of A100 hours, this leads to a cost of over **8000$ for a single trial** using cloud GPUs.
> > >
> > > Nonetheless, we've made every effort to showcase GET's scalability and parameter efficiency on CIFAR-10.
> > > In addition, we would like to do the code extension for ImageNet and seek collaborative community-drive efforts to execute large-scale distillation experiments.
> > >
> > > Thank you again for your valuable suggestions! We hope this discussion can further answer your questions.

---

> > > > ### Comment · Reviewer_GPxt · 2023-08-21
> > > >
> > > > Thanks for the clarifications!
> > > > I agree that computational cost should not be a limiting factor for publication. Therefore, I urge the authors to publish the training protocol and code so that the method can be explored by a broader community (I am particularly interested in distilling a pre-trained model like SDXL).
> > > >
> > > > All in all, I'm inclined to accept the work.

---

### Official Review · Reviewer_UV2u · 2023-07-13

**Soundness:** 3 good
**Presentation:** 3 good
**Contribution:** 2 fair
**Rating:** 5
**Confidence:** 4

**Summary:**

This paper proposes a new model architecture based on deep equilibrium models and applies it for distilling pretrain diffusion models, by generating samples offline and only utilizing the noise-sample pairs generated by the diffusion models. The architecture consists of embedding, injection transformer, equilibrium transformer and decoder. Empirical results show that the proposed approach leads to comparable performance to the other online distillation approaches, and the architecture is more efficient than ViT architecture in terms of training efficiency, model capacity and inference speed.

**Strengths:**

- The paper is clearly written and easy to follow.
- The paper shows that deep equilibrium model can be a competitive model class in the context of distilling diffusion models.
- Extensive comparison with ViT-based models demonstrates the effectiveness of the GET architecture.

**Weaknesses:**

- It's not clear to me why the work proposes to apply GET to this specific problem of distilling diffusion models. In principle, this architecture can serve as training a diffusion model directly similar to DiT. And I'm not convinced that offline distillation of diffusion models is a promising direction to work with, given the expensive sampling time the diffusion models need to generate the offline data (and that's why we need distillation). It would be more convincing if the work can show the effectiveness of the proposed architecture in a broader context, or explaining why the proposed architecture is especially useful for distillation.
- in l.228 it is claimed that other distillation approaches use much larger data budget but I don't think this comparison is fair enough, as the other approaches only require one functional evaluation for generating one sample, while this approach requires go through the whole sampling process of the diffusion model. I'd like to see the comparison of how many functional evaluations of the teacher model is required for different approaches.
-  I'd like to see the comparison of training/sampling speed with other distillation approaches as increasing sampling speed is the main motivation for doing distillation. I'm a bit concerned that the fixed point iteration would hinder the training/sampling speed compared to the original U-Net architecture.

**Questions:**

- How is z initialized at the start of fix point iteration. Is it sensitive to the initialization of z?
- In table 4, would be great to include the sampling speed as well.


**Limitations:**

Yes.

---

> ### Author Rebuttal · Authors · 2023-08-10
>
> Thank you for your careful review and good questions. Please find our responses to your questions and concerns below.
>
> __Motivation to use DEQ for distillation__: Our motivation to model the student network as a DEQ stems from the observation that the relatively complex process of distilling diffusion models has an element of fixed point process, as indicated by recent works like Consistency Models (Song et al. 2023) and TRACT (Berthelot et al. 2023). Ideally, we want a network architecture that has the ability to adapt its compute requirements to tradeoff perceptual quality for compute. DEQs possess both of these capabilities. The empirical results in our paper indeed indicate that modeling the student network (i.e. GET) as a DEQ is vital to achieve good performance in our set-up of one-step distillation.
>
> __Why offline distillation?__ Our primary goal is to demonstrate a relatively simple technique for distillation that does not use any trajectory information but can still achieve comparable performance to much more complex distillation techniques like progressive distillation. We generate data (noise/image pairs) in an offline manner to reduce the overhead due to sampling  during training. Further, this data generation needs to be done only once, and the generated  data can be reused later. Offline distillation with GET also requires fewer teacher evaluations compared to progressive distillation and consistency distillation, as we demonstrate in the next point.
>
> __Comparison of overall number of functional evaluations of teacher model__:
> Progressive distillation requires a significantly higher number of function evaluations for the teacher network than offline distillation. We state the exact number of function evaluations of teacher model below:
>
> **GET**: 1M (data samples) * 35 (NFEs for EDM sampling) = 35M
> **Progressive distillation** (assuming batch size of 128 for each of 8 TPUs):  2 DDIM steps * 128 (batch size) * 8 (TPUs) * (12 passes * 50k (iterations) + 100K (iterations for the last pass of 2 step to 1 step)) = 1.433B
> **Progressive distillation** (assuming batch size of 128 is shared across 8 TPUs):  2 DDIM steps * 128 (batch size) * (12 passes * 50k + 100K) = 179M
> **Consistency distillation**:  512 (batch size) * 800k (iterations)= 409.6M. In addition, the perceptual loss requires the same number of network evaluations as the teacher model.
>
> Note that progressive distillation uses DDIM with 8192 steps for CIFAR10.
>
> __Broader experiments with GET as a general architecture__: We acknowledge that GET is a general architecture that might have potential advantages in other applications. Our application of GET in distilling diffusion models was motivated by the inherent fixed point solving mechanism of the distillation process. We empirically demonstrated advantages of GET architecture in distillation over the non-weight tied counterparts. We leave the investigation of performance benefits of GET in other potential applications as future work.
>
> __GET as a backbone architecture for diffusion models__: While both GET and DiT are inspired by ViT, a primary architectural difference lies in the absence of temporal embeddings in GET which makes it tricky to use the current GET as a backbone architecture to train diffusion models from scratch. While it is feasible to incorporate temporal embeddings into GET and train it using denoising loss objective, conventional training of diffusion backbone architectures with this loss objective involves denoising images at a single time step. As a result, the advantages of the fixed point solving mechanism used in DEQs might be less pronounced in this case. In contrast, the process of distilling diffusion trajectories inherently involves a fixed point solving mechanism, which allows the benefits of fixed point solving mechanism to be better highlighted in this specific application.
>
> __Comparison of training and sampling speed with other distillation methods__:
>
> To ensure a fair comparison between GET and PD, we need to benchmark GET against PD baseline that uses ViT/DiT architecture. However, we do not have access to a ViT/DiT model that is distilled via PD on CIFAR-10. As a proxy for PD, we report the sampling time of EDM (50k images, 1NFE). A single forward pass through EDM requires 27s, whereas GET-Tiny completes sampling in 14.7s, GET-Mini takes 23.3s, and GET-Small needs 35.7s (all under 6 iterations).
>
> Considering that we use significantly fewer teacher model evaluations than PD (35M vs 179M (lower bound) or 1.433B (upper bound)), the performance can potentially be improved by using more training data and by increasing training iterations. As an example, the GET-Tiny with 8.9M parameters and 14.7s sampling time can reach FID of 11.47 and IS of 8.64 (while PD-EDM has IS of 8.69 at > 410M teacher evaluations), given 4$\times$ training iterations on *the same data* (35M teacher evaluations).
> In case, there is a publicly available model that the reviewer would like us to benchmark against, we are happy to evaluate it.
>
> > In Table 4, it would be great to include the sampling speed as well.
>
>  The sampling speed of ViT-B is 21.66 secs, and that of GET-Mini is 17.18 secs (4 iterations), 20.16 secs (5 iterations) and 23.27 secs (6 iterations), respectively. We will update Table 4 to include these sampling speeds.
>
> __Initialization of z at the start of fixed point iteration__ : We initialize z at the beginning of fixed point iterations by sampling it from the standard Gaussian distribution. We expect the fixed point iterations to show limited sensitivity to the initial value of z, provided that this value is reasonably set.
>
> Please reach out to us in case you have any follow up questions/concerns.

---

> > ### Comment · Reviewer_UV2u · 2023-08-15
> >
> > Thank the authors for addressing my questions in the rebuttal. I decided to increase my score from 4 to 5, mainly due to the clarification of motivation, and the teacher NFE comparison with existing distillation approaches.

---

### Official Review · Reviewer_BXuM · 2023-07-18

**Soundness:** 3 good
**Presentation:** 3 good
**Contribution:** 2 fair
**Rating:** 6
**Confidence:** 4

**Summary:**

Distillation of diffusion models into smaller models that require fewer steps for generation is an important topic in current research. The authors propose to use a deep equilibrium model as the student. In particular, the student model is a Generative Equilibrium Transformer (GET) that consists of two main modules -- (a) an Injection Transformer that outputs a noise embedding and the (b) Equilibrium Transformer that is trained using the fixed point iteration where the fixed point is the latent embedding of a clean image, which when passed through an image decoder outputs the image output by the diffusion model. GET is trained using a simple L1 loss between the output of GET and the image generated by the diffusion model. At test time, a few steps of performing forward passes through the equilibrium transformer seem sufficient to generate a clean image. The paper also performs a series of experiment demonstrating advantages in terms of data and parameter efficiency, sampling speed etc. compared to existing methods.

**Strengths:**

1. The paper is easy to understand at the implementation level and well-written.

2. The authors perform a good set of experiments to understand the trade-offs between data and parameter efficiency, sampling speed, training compute and generation quality. These experiments put the proposed method favorably compared to existing methods.

3.  In particular, with only about three forward passes through the distilled model, with a fraction of the parameters and sampling time compared to a ViT, better FID scores are observed using the proposed method.

**Weaknesses:**

1. I am not sure if the proposed method can be called one-step generation. My understanding is that some steps through the equilibrium transformer are needed before getting close to convergence to the fixed point. This is also shown in Figure 3. Perhaps some clarification is needed as to what the one-step means in the title. The equilibrium model is trained to produce an image from noise directly, but it still needs multiple passes to get a good quality image.

2. CIFAR-10 is used for the experiments. While the generated images look okay, it is hard to determine their quality as they are small. Evaluation on just CIFAR-10 is weak in my opinion. A dataset with larger images should make the results stronger.

3. Some of the design choices seem arbitrary. For example, why is a separate Injection Transformer necessary? Why the student model has to be a Deep Equilibrium Model is also not clear from the paper. What about training to produce the fixed point is important for distillation? These questions can be partly answered with ablation analysis, but some intuition should also be provided.



**Questions:**

Please see weaknesses above.

**Limitations:**

Yes, limitations have been adequately addressed.

---

> ### Author Rebuttal · Authors · 2023-08-10
>
> Thank you for your encouraging feedback! We have tried our best to address all your questions and concerns below.
>
> __Clarification about one-step generation__: We define one-step generation as the ability to generate an image directly from Gaussian noise in a _single_ forward pass through the network.  There can be multiple iterations over the equilibrium transformer during this forward pass but it is important to note that all of these iterations contribute to the depth of the network. In Figure 3(a), we demonstrate how the performance changes with variation in the effective depth of GET. Standard neural network architectures (e.g., ResNet, Diffusion Transformer (DiT) ) increase the depth of the network by applying a block multiple times, increasing the number of parameters in the process. In contrast, GET weight-ties the repeated block, preserving the overall number of parameters.
>
> __Evaluation on larger datasets__: We agree that large scale experiments will provide more comprehensive insights into the capabilities of GET. However, scaling up to ImageNet would require significantly more computing resources. For comparison, consistency distillation uses 64 Nvidia A100 GPUs for experiments on ImageNet-64X64 and LSUN 256X256. Progressive Distillation reports use of 64 TPUv4 chips for their large scale experiments on ImageNet-64X64 and LSUN 256X256.
>
> __Clarification about design choices within GET__:
>
> - __Necessity of the Injection Transformer__: The reviewer has a valid concern about the role of the injection transformer. To address this concern, we trained a model that does not have the injection transformer, but is otherwise architecturally similar to GET. We report the results in the table below.
>
> | Model | Params | FLOPS | FID  |  IS  |
> |----------|:-------------:|:---:|:----:|:----:|
> | GET-M   	  | 19.2M  | 15.2G  | 10.72 | 8.69 |
> | GET-M-no-inj  |  10.7M | 15.5G  | 14.10 | 8.55 |
>
>
> Thus the injection transformer improves GET’s performance.
>
> - __Motivation of the student model to be a DEQ__: Our motivation to model the student network as a DEQ stems from the observation that the relatively complex process of distilling diffusion models has an element of fixed point process, as indicated by recent works like Consistency Models (Song et al. 2023) and TRACT (Berthelot et al. 2023). Ideally, we want a network architecture that has the ability to adapt its compute requirements to tradeoff perceptual quality for compute. DEQs possess both of these capabilities. The empirical results in our paper indeed indicate that modeling the student network (i.e. GET) as a DEQ is vital to achieve good performance in our set-up of one-step distillation.
> We also include new empirical results that we report in the table below which show that a weight-tied GET outperforms the non-weight tied counterpart in terms of both FID and IS, despite having significantly more parameters. We note that this non-weight-tied model has the same FLOPs as the original weight-tied model. This outcome further underscores that weight-tying within GET indeed contributes to improved performance.
>
> | Model | Params |  FID  |  IS  |
> |----------|:-------------:|:----:| :----:|
> | GET-B   	    | 62.2M | 7.42 | 9.16 |
> | GET-B-non-WT | 310M  | 7.55 | 9.08 |
>
> - __Why train for fixed point iterations?__ As mentioned in the previous point, modeling GET as a DEQ that solves for the fixed point helps to achieve superior one-step distillation compared to its non-weight tied counterpart. Figure 3 in the main paper also corroborates this, as it indicates that fixed point iterations lead to improved FID scores.
>
> We are happy to answer any follow up questions and address any concerns you may have.

---

> > ### Comment · Reviewer_BXuM · 2023-08-18
> > **Thank you for the response**
> >
> > I thank the authors for their response.
> >
> > I acknowledge that I have read their response to my comments as well as those of the other reviewers.
> >
> > I think what the authors mean by one-step generation is now clear.
> >
> > As for some important design choices -- the injection transformer and the student model being a deep equilibrium model -- the ablation experiments are certainly helpful and appreciated. However, the intuitions still remain unclear and tenuous.
> >
> > The NFE comparisons with other competing models is also a very good result.
> >
> > Overall, the paper has some good ideas and good results, so I am recommending a Weak Accept.

---

### Author Rebuttal · Authors · 2023-08-10

We thank all the reviewers for their thoughtful comments and suggestions. Here, we address some of the common concerns raised by the reviewers.

__What does one-step generation mean?__ We define one-step generation as the ability to generate an image directly from Gaussian noise in a _single_ forward pass through the network. While there can be multiple iterations within the equilibrium transformer in this single forward pass, all of these iterations contribute to the “effective depth” of the network.

__Evaluation on larger datasets__: We agree with the reviewers that experiments on larger datasets like ImageNet will significantly strengthen the results. However, scaling up to ImageNet would require significantly more computing resources. For comparison, consistency distillation uses 64 Nvidia A100 GPUs for experiments on ImageNet-64$\times$64 and LSUN 256$\times$256. Progressive distillation needs 5 days on 64 TPUv4 chips for experiments on ImageNet-64$\times$64 and LSUN 256$\times$256. This is equivalent to/more than 7680 A100 GPU hours.  We have tried our best to demonstrate the scaling capabilities of GET in terms of training FLOPs and parameters (See Figure 3a/b) to support the possibility of transferring GET to more complex settings.

__New result on teacher network evaluations__: Offline distillation with GET requires significantly fewer teacher model evaluations (35M) compared to progressive distillation (179M (lower bound) or 1.433B (upper bound)) and consistency distillation (409.6M).

---

### Decision · Program_Chairs · 2023-09-21

**Decision:**

Accept (poster)

**Comment:**

This paper proposes a Deep Equilibrium Models (DEM) based architecture for offline one-step distillation of diffusion models. Experimentally, it is shown that such an architecture is favorable compared to a standard ViT, and other distillation methods such as PD. The paper has received an average rating of 5.33, which is a borderline score, but all reviewers are leaning towards acceptance. The AC agrees that there is merit to the proposed method, but the paper could be further strengthened with more evaluations as suggested by the reviewers. In addition to what's being discussed by the reviewers, the AC believes that it would be beneficial to conduct a direct comparison to knowledge distillation (Luhman & Luhman, 2021), while evaluating both with the same teacher model. The AC recommend accepting the paper, but strongly encourage the authors to incorporate the suggestions from the reviewers in the final version.